# Deep learning-driven insights into super protein complexes for outer membrane protein biogenesis in bacteria

**Mu Gao[1]\*, Davi Nakajima An[2], Jeffrey Skolnick[1]\***

[1]Center for the Study of Systems Biology, School of Biological Sciences, Georgia Institute of Technology, Atlanta, United States; [2]School of Computer Science, Georgia Institute of Technology, Atlanta, United States

**Abstract** To reach their final destinations, outer membrane proteins (OMPs) of gram-negative bacteria undertake an eventful journey beginning in the cytosol. Multiple molecular machines, chaperones, proteases, and other enzymes facilitate the translocation and assembly of OMPs. These helpers usually associate, often transiently, forming large protein assemblies. They are not well understood due to experimental challenges in capturing and characterizing protein-protein interactions (PPIs), especially transient ones. Using AF2Complex, we introduce a high-throughput, deep learning pipeline to identify PPIs within the *Escherichia coli* cell envelope and apply it to several proteins from an OMP biogenesis pathway. Among the top confident hits obtained from screening ~1500 envelope proteins, we find not only expected interactions but also unexpected ones with profound implications. Subsequently, we predict atomic structures for these protein complexes. These structures, typically of high confidence, explain experimental observations and lead to mechanistic hypotheses for how a chaperone assists a nascent, precursor OMP emerging from a translocon, how another chaperone prevents it from aggregating and docks to a β-barrel assembly port, and how a protease performs quality control. This work presents a general strategy for investigating biological pathways by using structural insights gained from deep learning-based predictions.

**\*For correspondence:**
mu.gao@gatech.edu (MG);
skolnick@gatech.edu (JS)

**Competing interest:** The authors declare that no competing interests exist.

## Editor's evaluation

The authors show that an artificial-intelligence method can be used to predict the three-dimensional structure of protein-protein complexes formed between cellular factors that promote the assembly of bacterial outer membrane proteins. The structures are compelling because they explain previously published biochemical data and provide novel insights into the function of these factors.

## Introduction

A structural component unique to gram-negative bacteria is the outer membrane (OM), composed of lipopolysaccharides and phospholipids in an asymmetric bilayer with embedded lipoproteins and transmembrane β-barrel proteins (*Silhavy et al., 2010*). The latter group, OM proteins (OMPs), play vital functional roles, e.g., exchanging small molecules with the environment through their transmembrane β-barrel porins. OMPs are synthesized by cytosolic ribosomes, translocated across the inner membrane (IM), and finally delivered to the OM via the OMP biogenesis pathway (*Rollauer et al., 2015*).

The translocation and folding of OMPs involve many proteins, including essential and auxiliary ones that form multiple complexes in cooperation (*De Geyter et al., 2016*; *Troman and Collinson,*

**eLife digest** All living cells are contained within a fatty cell membrane that allows water and only certain other molecules to pass through with ease. Bacteria only consist of a single cell, making their membrane the only interface with the surrounding environment. Gram-negative bacteria – which include *Escherichia coli*, a bacterium found in the gut of all humans – have an extra layer of protection, the 'outer membrane'. Proteins in this membrane are called 'outer membrane proteins' or OMPs and allow nutrients to enter the cell. But OMPs, which are made inside the cell, need to be transported to the outer membrane and folded correctly before they can perform their role. This multistep process, which involves interactions between many different proteins, is not fully understood.

The journey of an OMP from the center of the cell where it is made to the outer membrane is complicated. First, the OMP needs to pass through the cell's inner membrane. To do this, it must interact with 'channel proteins' in the inner membrane that feed the OMP into the space between the two membranes, known as the bacterial envelope. This step requires the OMP to be unfolded**.** Once in the bacterial envelope the OMP interacts with proteins that help it fold correctly and integrate into the outer membrane.

The interactions between proteins in the bacterial envelope are short-lived, making them difficult to study using lab-based experiments. An alternative approach is predicting a protein's structure from its amino acid sequence which is a difficult computational problem to solve. However, in 2020 developers behind the AlphaFold2, a deep learning program, were able to use a set of equations organized in a 'neural network' that can 'learn' from a library of known protein structures to predict unknown structures with high accuracy. Gao et al. used AF2Complex, a tool based AlphaFold2, tailored to predicting interactions between proteins, to investigate what interactions OMPs could be involved with on their way to the outer membrane.

With the help of a supercomputer at the Oakridge National Laboratory, Gao et al. screened nearly 1,500 *E. coli* proteins within the bacterial envelope to see how they might interact with OMPs. The screen identified previously unknown interactions between proteins that suggest that the formation of the bacterial outer membrane and the integration of proteins into it involve protein complexes and molecular mechanisms that have not yet been characterized. Additionally, the screen also identified interactions that had been previously described, confirming that the deep learning approach can correctly capture real interactions.

Overall, Gao et al.'s work inspires new hypotheses about the mechanisms through which OMPs are transported to the outer membrane, although further work will be needed to confirm the roles of protein interactions predicted by the computational model experimentally. Furthermore, the ability to design experiments based on computational predictions is exciting. If confirmed, the new protein interactions could help scientists better understand OMP transport, which is essential for bacterial biology. In the future, this could lead to the discovery of new targets for antibiotic drugs.

---

*2021*). Two core complexes are the SecYEG translocon (*Rapoport et al., 2017*; *Oswald et al., 2021*) and the β-barrel assembly machine (BAM; *Tomasek and Kahne, 2021*; *Noinaj et al., 2017*). SecYEG, a hetero-trimer composed of the secretion channel SecY and two additional subunits SecE and SecG, is anchored to the IM and is responsible for moving most cell envelope proteins across the IM (*Rapoport et al., 2017*). Periplasmic and OM proteins enter the channel in SecY via the SecA-dependent translocation pathway (*Oswald et al., 2021*). They possess signal peptides recognized by SecA, which inserts a protein substrate into SecY and powers it through the channel using energy from ATP hydrolysis (*Collinson, 2019*). The signal peptide, located at the N-terminus of a substrate, contains a hydrophobic segment that folds into a transmembrane α-helix once it exits the lateral gate of SecY. To release the substrate from the IM, a type I signal peptidase (SPase I) cleaves the signal peptide (*Paetzel, 2014*). At this point, a periplasmic protein has reached its destination, but an OMP, escorted by the chaperones SurA or Skp, continues its journey toward BAM harbored in the OM (*Sklar et al., 2007*). Five subunits constitute BAM, of which BamA plays the major role in folding a β-barrel and releasing the matured product (*Tomasek and Kahne, 2021*; *Noinaj et al., 2017*).

There are many open questions concerning the OMP biogenesis pathway. In *Escherichia coli*, the SecYEG translocon recruits additional proteins, such as SecA (*Li et al., 2016*), SecDF (*Tsukazaki,*

*2018*), YidC (*Kumazaki et al., 2014*), and PpiD (*Antonoaea et al., 2008*; *Sachelaru et al., 2014*), to form a variety of supercomplexes in different scenarios. We do not know the identities of all members of these supercomplexes, let alone their atomic structures. SurA plays a major role in chaperoning a nascent OMP (*Sklar et al., 2007*; *Behrens-Kneip, 2010*). How does it handle and deliver a substrate to BAM? Likewise, BAM recruits additional helpers, e.g., BepA (*Narita et al., 2013*), but how do they work together?

To answer these questions experimentally is challenging (*Babu et al., 2018*; *Maddalo et al., 2011*; *Alvira et al., 2020*; *Carlson et al., 2019*). Recently, deep learning approaches have made tremendous progress in predicting the structures of protein complexes (*Gao et al., 2022b*; *Humphreys et al., 2021*; *Evans et al., 2021*; *Bryant et al., 2022*). Here, we use one such method to address the above questions. The centerpiece of our approach is AF2Complex (*Gao et al., 2022b*), built on AlphaFold2 (AF2) (*Evans et al., 2021*; *Jumper et al., 2021*). Using AF2Complex, we combine virtual screening for protein-protein interactions (PPIs) and supercomplex modeling and apply this strategy to several important proteins in the OMP biogenesis pathway.

## Results
### Virtual screening for PPIs in the *E. coli* envelopome

A workflow employing AF2Complex was implemented to search for interacting proteins within the *E. coli* cell envelope. We refer to these proteins collectively as the 'envelopome,' which consists of ~1450 proteins, or ~35% of the complete *E. coli* proteome. Given a query envelope protein sequence, screening is conducted against every envelope protein, including the query itself to test if it forms a homo-oligomer. For each pair of input proteins, AF2Complex predicts their structures simultaneously as a putative dimer and evaluates the interface score (iScore) of the resulting structural models to decide if they interact. The value of iScore ranges between 0 and 1. Interacting protein pairs typically return a positive iScore, whereas non-interacting proteins usually return 0 or low iScores. Typically, 20 final structures are produced in multiple independent runs using 10 different AF2 deep learning models. The highest iScore among all predicted structures of a target is used for ranking the entire envelopome.

According to previous benchmarks on a set of ~7000 putatively non-interacting protein pairs from *E. coli*, minimum iScore thresholds of 0.40, 0.50, and 0.70 yield false positive rates of 1.2, 0.4, and <0.01%, respectively (*Gao et al., 2022b*). With respect to the capability of identifying and modeling true PPIs, on a set of 440 heterodimeric complexes whose experimental structures have been recently determined and were not used for AF2 model training, AF2Complex recalls 81, 74, and 34% of these benchmark targets at the same three iScore thresholds, respectively, and yields medium- or high-quality complex structures for 84, 87, and 93% in the top ranked models of these positively identified targets (*Gao et al., 2022b*). Hence, we consider predictions of medium, high, and very high confidence, progressively, at iScore cutoffs of 0.4, 0.5, and 0.7. Given a query protein, the top iScores of all scanned envelope proteins are then sorted to identify potential interacting partners with the query protein. Note that our computational predictions are about physical interactions between a pair of proteins subjected to screening, not about their biological roles even if they are predicted to interact physically. Moreover, the predicted physical interactions may not be relevant in a cellular environment due to various factors not considered in modeling, e.g., competition from other proteins with stronger binding affinities, post-translational modifications, etc. Thus, it is possible that many PPIs predicted by this pipeline do not necessarily have biological relevance. Nevertheless, since cognate PPIs required by their functions are more likely to be detected than randomly selected proteins, biologically interesting PPIs are enriched at the top of the screening results ranked by iScore. Thus, the screening procedure may provide valuable even critical clues for subsequent investigation. In this study, assisted by existing experimental evidence, we select from high confidence computational predictions those most likely to have significant biological implications and then predict the structures of larger complexes if more than two proteins are involved according to our predictions or based on literature information. The interactions that we ignored are either of unknown biological significance, physically interacting but biologically irrelevant, or simply false positives.

As a first test, we applied our PPI pipeline to the chaperones PpiD and YfgM because strong experimental evidence indicates that these two proteins belong to a super SecYEG translocon (*Antonoaea*

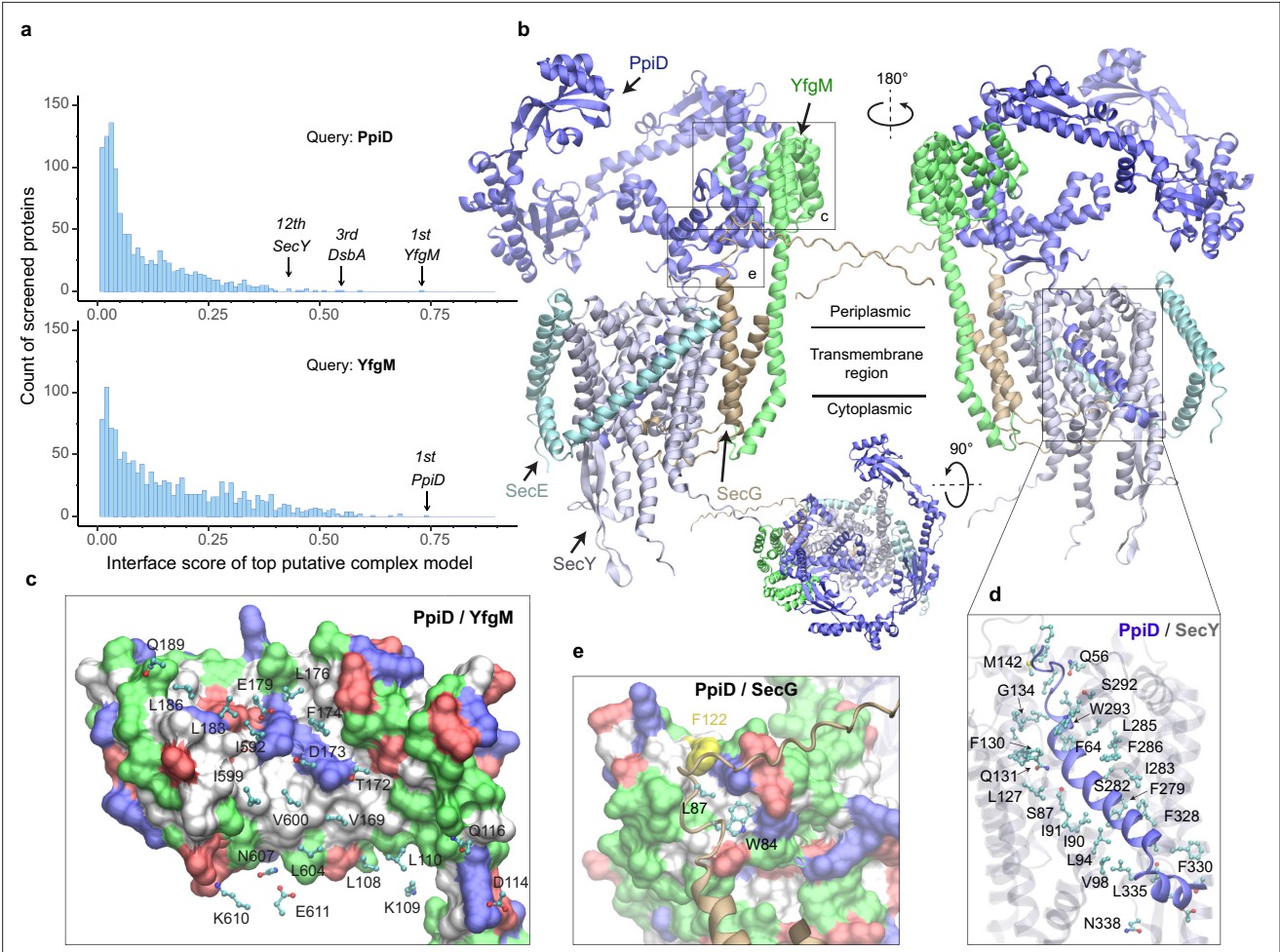

**Figure 1.** *E. coli* super-translocon SecYEG/PpiD/YfgM. (**a**) Computational screening for protein-protein interaction partners of PpiD and YfgM within the *E. coli* envelopome, respectively. A histogram displays the distribution of the top interface scores (iScores) of all envelope proteins screened with each query. Black arrows mark the top hits that were further studied, along with their names and overall ranks. (**b**) The top AF2Complex model of a supercomplex made of PpiD (blue), YfgM (green), SecY (silver), SecE (cyan), and SecG (tan) in three different views. Proteins are shown in a cartoon representation. Viewpoint transition, from either left to right or top to bottom, is indicated by a rotation axis (dashed line) and the rotation angle in degrees (circled arrow). (**c, d, and e**) Predicted PPI sites. The corresponding locations in **b** are indicated by black boxes. For clarity, the viewpoints and representations are adjusted. In the surface representations **c** and **e**, the color code is hydrophobic (white), polar (green), positive (blue), and negative (red), except for Phe122$_{PpiD}$ (yellow) in **e**. The same color code for the surface representation is employed below unless noted otherwise. PPI residues are shown in a ball-and-stick representation for PpiD in **c**, SecY in **d**, and SecG in **e**; the color scheme of atoms is carbon (cyan), oxygen (red), nitrogen (blue), and sulfur (yellow). The same scheme of atoms is adopted throughout this work.

The online version of this article includes the following figure supplement(s) for figure 1:

**Figure supplement 1.** Comparison of a computed structure of SecY (silver) and two experimental structures (magenta).

---

*et al., 2008*; *Sachelaru et al., 2014*; *Maddalo et al., 2011*; *Dartigalongue and Raina, 1998*; *Fürst et al., 2018*; *Jauss et al., 2019*; *Matern et al., 2010*; *Götzke et al., 2014*), yet their structures remain elusive. *Figure 1a* shows the screening results of PpiD and YfgM against the envelopome. Reassuringly, these two proteins stand out at the very top of the predicted PPI rankings among the *E. coli* envelopome with iScores of ~0.73 in each case (*Figure 1a*, *Supplementary file 1*). Furthermore, PpiD hits SecY at the 12th rank (iScore = 0.43), corroborating previous experimental studies that SecY is an interacting partner of PpiD (*Antonoaea et al., 2008*; *Sachelaru et al., 2014*; *Fürst et al., 2018*).

## SecYEG associates with the chaperone PpiD and assistant YfgM

Subsequently, we modeled SecYEG together with PpiD and YfgM and acquired another very high confidence structure (iScore = 0.73, mean predicted local distance difference test [pLDDT] = 79,

see Methods; *Figure 1b–e*). Between this supercomplex and the dimer structures obtained from screening, the PPI interfaces between PpiD and YfgM are very similar according to iAlign (*Gao and Skolnick, 2010*), which yields a root-mean-square deviation (RMSD) of 0.6 Å for the $C_\alpha$ atoms of the interface residues. Thus, we focus on the supercomplex structure below.

Individually, PpiD and YfgM in our complex model display the same architecture as their respective monomer models calculated by applying AF2 (*Varadi et al., 2022*). PpiD by itself has four domains in an open-arm shape, fitting its proposed chaperone role (*Antonoaea et al., 2008*; *Matern et al., 2010*), plus an N-terminal transmembrane α-helix (*Figure 1b*). The membrane anchoring helix is linked to a domain consisting of two discontinuous segments (residues 40–189, 581–623) from the terminal regions. Within this domain, two α-helices (residues 167–189, 581–609) are held by the tetratricopeptide repeat (TPR) domain of YfgM, which inserts itself into the IM via a single hydrophobic α-helix. Structurally, the PpiD/YfgM interface resembles a pair of 'chopsticks' (two PpiD helices) grasped by a 'palm' (the TPR domain). Their PPI interactions are quite extensive, including ~100 inter-protein residue-residue contacts involving both hydrophobic and polar partners (*Figure 1b and c*), which may explain the strong binding affinity observed in experiments (*Maddalo et al., 2011*).

In addition to the channel that connects the cytosol to the periplasm, SecY features a lateral gate embedded in the IM. It opens to release transmembrane segments, including the signal peptide of a substrate such as OmpA (*Li et al., 2016*). In vivo photo cross-linking studies found numerous sites at the lateral gate probed by PpiD (*Sachelaru et al., 2014*; *Jauss et al., 2019*). Such cross-linking suggests direct physical contacts between the two proteins because the covalent linkers formed upon UV light radiation are very short, lying within 4 Å. Because PpiD has only a single transmembrane α-helix, a logical speculation is that the α-helix interacts with the lateral gate of SecY. Indeed, our structure reveals that PpiD guards the lateral gate of SecY with its α-helix (*Figure 1b and d*), which occupies the same position as a signal peptide leaving the SecY gate (*Figure 1—figure supplement 1*). In superposition with a SecY crystal structure co-crystallized with an OmpA signal peptide, the $C_\alpha$ RMSD of SecY is 2.4 Å between the X-ray and computed structures. Thus, SecY in our model is open to accommodate the α-helix of PpiD, in comparison to an electron microscopy (EM) structure of *E. coli* SecY in which the gate is closed (*Figure 1—figure supplement 1*). Our model further elucidates why SecY residues located in transmembrane helices (TM2/3/7/8) can be cross-linked to PpiD, because they are located at the PpiD/SecY interface. For example, strong cross-linking signals have been found for five SecY residues (*Sachelaru et al., 2014*; *Jauss et al., 2019*), Ile91$_{SecY}$, Leu94$_{SecY}$,

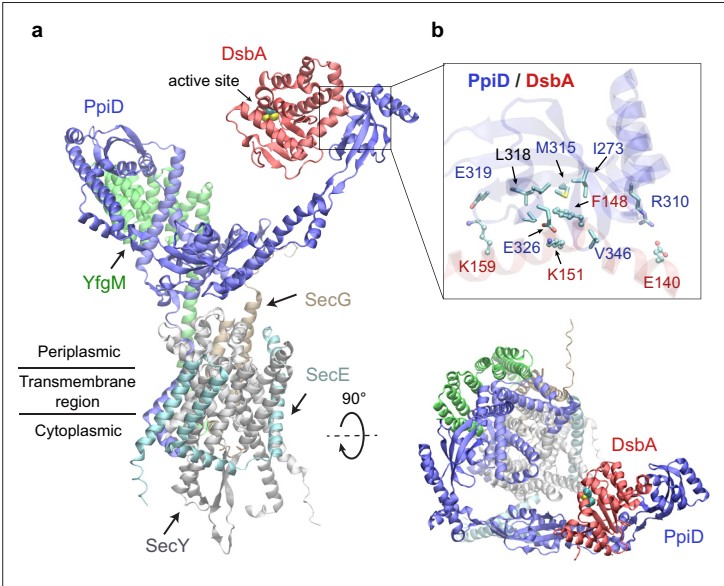

**Figure 2.** Structural model of the SecYEG/PpiD/YfgM/DsbA supercomplex. (**a**) Two views of the predicted structure. DsbA is shown in red, while the other proteins are colored the same as in *Figure 1*. Two cysteines, Cys49 and Cys52, essential to the enzymatic function of DsbA, are shown as spheres. (**b**) Protein-protein interaction sites between PpiD and DsbA. For clarity, tertiary structures are transparent. Key interacting residues are shown in the licorice representation for PpiD and in the ball-and-stick representation for DsbA.

Leu127$_{SecY}$, Phe130$_{SecY}$, and Phe286$_{SecY}$, and all make direct physical contacts (<4.5 Å) with PpiD in our structural model (*Figure 1d*). Moreover, we identified direct contacts between a hydrophobic pocket of PpiD and SecG, explaining the experimentally observed crosslinking of Phe122$_{PpiD}$ to SecG (*Fürst et al., 2018*; *Figure 1e*). The interaction assigns SecG a functional role in coordinating with PpiD. Overall, the predicted structure rationalizes the results of several experimental studies.

## The chaperone PpiD interacts with the disulfide isomerase DsbA

Among the top hits for PPI partners of PpiD, a fascinating discovery is DsbA, ranked third with a high confidence iScore of 0.55. DsbA donates its disulfide bond to a substrate in need; thus, it is critical to the folding of a nascent protein as it leaves the SecY channel (*Kadokura and Beckwith, 2009*; *Goemans et al., 2014*). How does DsbA coordinate with the SecY translocon? The predicted interaction immediately provides an answer, i.e., by associating with the chaperone PpiD that is a part of a super translocon complex (SecYEG/PpiD/YfgM/DsbA). In a predicted structure of this supercomplex (iScore = 0.71, mean pLDDT = 78, *Figure 2a*), PpiD uses its 'hand' to grip DsbA, as observed in the top model of the pair because their interactions do not interfere with other members of the translocon. The hand of PpiD is a parvulin-like domain, but it is devoid of the peptidyl-prolyl isomerase (PPIase) activity canonical to parvulin, due to the mutation of a critical histidine (*Weininger et al., 2010*). As our structure reveals, the catalytically inactive pocket of the PPIase domain is hydrophobic and buries the aromatic sidechain of Phe148$_{DsbA}$, which becomes a hot-spot residue surrounded by six hydrophobic residues of PpiD upon complexation (*Figure 2b*). In addition, three pairs of salt-bridges are formed around the hydrophobic contacts, contributing to specific recognition. Notably, the catalytic cysteines of DsbA, Cys49$_{DsbA}$ and Cys52$_{DsbA}$, are on the opposite side to the protein-protein interface (*Figure 2a*); thus, they are open to engage a substrate as it emerges from the SecY channel. Based on this model, we hypothesize that DsbA resides on PpiD transiently to improve its chance of encountering a substrate.

## SPase I LepB accesses an OMP substrate received by PpiD/YfgM

Polypeptide translocation triggers the dissociation of PpiD from SecY (*Sachelaru et al., 2014*). The modeled complex suggests that PpiD dissociation is realized by pushing a substrate out of the SecY gate, e.g., the signal peptide, which then repels the PpiD helix bound to the gate. For an OMP, a key next step is the removal of the signal peptide by the peptidase LepB in *E. coli* (*Paetzel, 2014*). How does it operate? The dissociation of PpiD from SecYEG vacates a space for LepB to approach a substrate. We generated a structural model of LepB in complex with a precursor OmpA (proOmpA) chain (residue 1–87) grasped by PpiD (iScore = 0.64, mean pLDDT = 80; *Figure 3*). The OmpA cleavage site, Ala21$_{OmpA}$, directly contacts the LepB active site triad, Ser89$_{LepB}$, Ser91$_{LepB}$, and Lys146$_{LepB}$ at ~4 Å

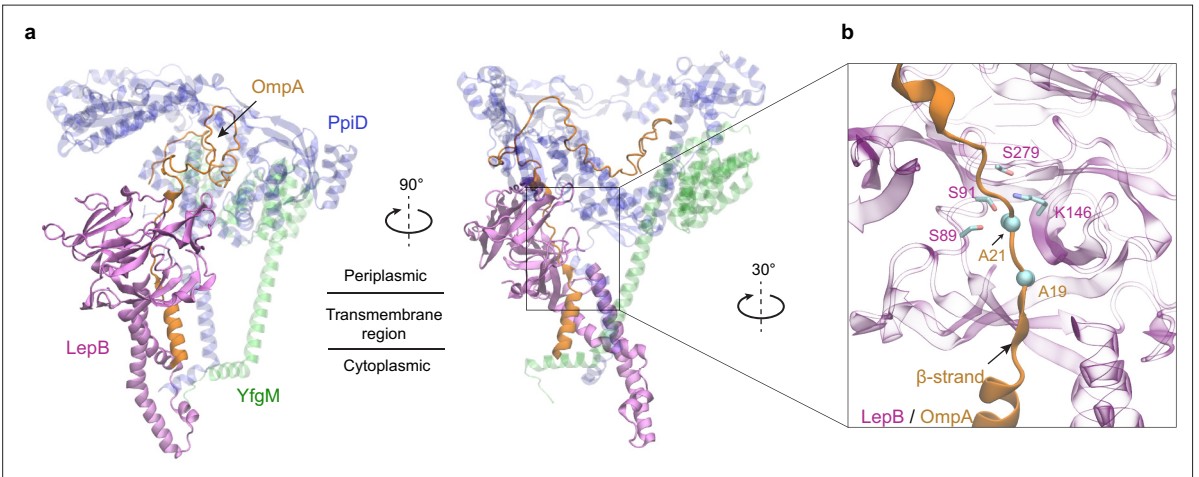

**Figure 3.** Predicted structure of the PpiD/YfgM/LepB/OmpA supercomplex. (**a**) Two views in the cartoon representation are shown. Colors: PpiD (blue), YfgM (green), LepB (magenta), and OmpA (residue 1–87, yellow). For clarity, representations of PpiD and LepB are transparent. (**b**) Close-up view of the OmpA signal peptide in the active site of LepB. Essential catalytic residues, Ser89, Ser91, Lys146, and Ser279 of LepB are shown in a licorice representation, and the cleavage site Ala21 and Ala19 of OmpA is shown as spheres.

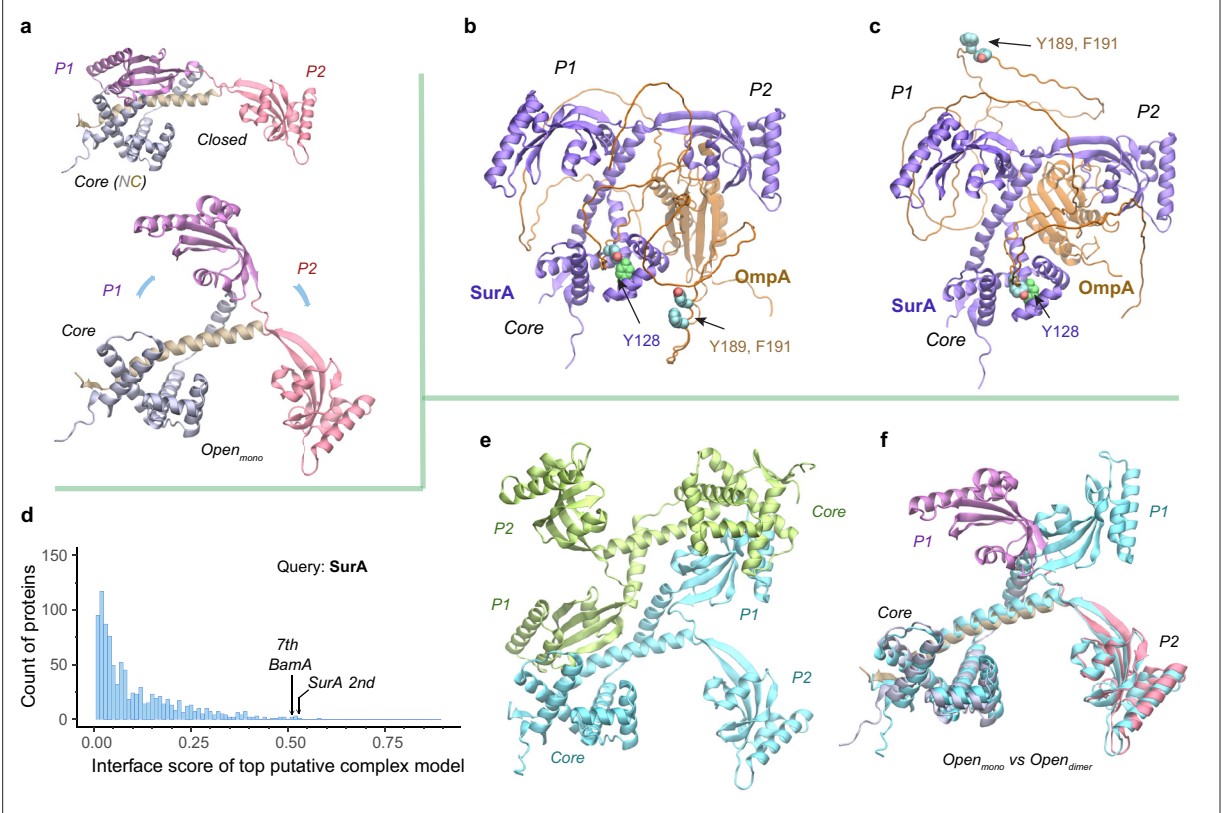

**Figure 4.** Structural models of SurA in the absence and presence of an OmpA substrate. (**a**) Open and closed conformations of monomeric SurA, consisting of the core domain (N-terminal region in gray and C-terminal in tan), P1 (purple), P2 (red). (**b and c**) Two structures of SurA in the presence of an OmpA substrate from two separate modeling runs. In both, SurA is open as in **a**. The β-barrel domain of OmpA is completely unfolded and generally does not maintain the same residue-residue contacts with SurA, except that an OmpA aromatic residue consistently makes π-π interactions with Tyr128$_{SurA}$ located in the crevice of the SurA core domain. Two β-signal residues, Y189 and F191 of OmpA, are also shown as spheres. The folded periplasmic domain of OmpA is bound to the P2 domain of SurA. (**d**) Envelopome protein-protein interaction screening of SurA identifies itself and BamA among the top hits. (**e**) Two views of the top predicted structure of a SurA dimer (green and cyan). (**f**) Superimposition of two open conformations from the monomeric and dimeric SurA. Subscripts indicate the stoichiometry. The color schemes correspond to those used in **a** and **e**. Only a single SurA from the dimeric model is shown in the superposition.

The online version of this article includes the following figure supplement(s) for figure 4:

**Figure supplement 1.** Structural models of the OmpA polypeptide in the absence of SurA.

away from the alanine. In our model, the proOmpA chain contributes a β-strand to form a parallel β-sheet with LepB, as has been speculated (*Paetzel, 2014*). Overall, the structure provides a model for how LepB accesses a substrate received by PpiD/YfgM to cleave a signal peptide.

## The chaperone SurA opens to load an OMP substrate

After passage through a SecYEG translocon, a nascent OMP polypeptide is relayed to SurA, which convoys the substrate toward its next stop, BAM. A crystal structure of SurA exhibits three domains: a core domain (split into N- and C-terminal subdomains) and two PPIase domains, P1 and P2 (*Bitto and McKay, 2002*). P2 retains PPIase activity but P1 does not (*Behrens et al., 2001*). In this crystal structure, P1 further packs with the core domain forming a cradle-like structure (*Bitto and McKay, 2002*), referred to as the closed conformation of SurA. By contrast, another crystal structure of the SurA homodimer lacking P2s displays an open conformation, where two P1s are uncoupled from the core domains to hold a dodecapeptide at the dimeric interface between P1s (*Xu et al., 2007*). To explore conformations of SurA, we predicted structures for a single, full-length SurA and obtain two major conformations (*Figure 4a*). One conformation closely resembles the cradle-like closed conformation with a backbone C$_{\alpha}$ RMSD of 2.4 Å from the crystal structure. The other mimics the open conformation but incorporates P2 absent in the X-ray structure. Overall, the open conformation resembles

a 'three-prong hook,' where both P1 and P2 swing away from their resting position in the closed conformation.

SurA functions mainly as a monomer but may act as a dimer for a large client (*Li et al., 2018*; *Calabrese et al., 2020*). How does SurA chaperone a substrate such as OmpA? OmpA consists of a β-barrel (N-terminal) (*Pautsch and Schulz, 1998*) and a periplasmic (C-terminal) domain (*Ishida et al., 2014*). To model the OmpA polypeptide, we reduced the size of multiple sequence alignments provided to AF2Complex and removed all its structural templates. Consequently, we generated a monomeric OmpA structure with a non-native, collapsed N-terminal domain and a native-like periplasmic domain (*Figure 4—figure supplement 1*). In the presence of SurA, the periplasmic domain maintains the same fold, but remarkably, the non-native β-barrel region completely unravels and wraps around SurA (*Figure 4b–c*). This is consistent with small angle neutron scattering data that suggests an expanded and unfolded OmpA substrate (*Marx et al., 2020*). The SurA/OmpA dimer is not stable as different wrapping configurations were observed in two separate structure models, leading to very low iScores of ~0.05, reflecting uncertainty due to the possibility of many conformations. Nevertheless, the SurA/OmpA models appear physical and provide a hypothetical basis for how the chaperone SurA could prevent a polypeptide chain from aggregating and present an unfolded polypeptide to BAM for its final assembly. Intriguingly, in both predicted complex structures, the OmpA polypeptide passes through a SurA crevice at its core domain, where $Tyr126_{SurA}$ consistently attracts a substrate aromatic residue with π-π interactions. This may be one mechanism by which SurA recognizes OMPs, which are enriched with aromatic residues (*Xu et al., 2007*; *Hennecke et al., 2005*). Moreover, the disordered β-barrel region has few interactions with SurA could explain how the system avoids high-energy expenditure for β-barrel release. Of note, two key 'β-signal' residues (*Tommassen, 2010*), Y189 and F191 of OmpA, do not make direct physical contact with SurA in both models.

To identify other interacting partners, we searched the *E. coli* envelopome for PPIs with SurA. Interestingly, SurA recognizes itself at a high confidence iScore of 0.53 (*Figure 4d*). The top model of a SurA homodimer exhibits twofold rotational symmetry, in which P1 domains are swapped, leading to an open conformation quite different from the above one (*Figure 4e–f*). This conformation, like a

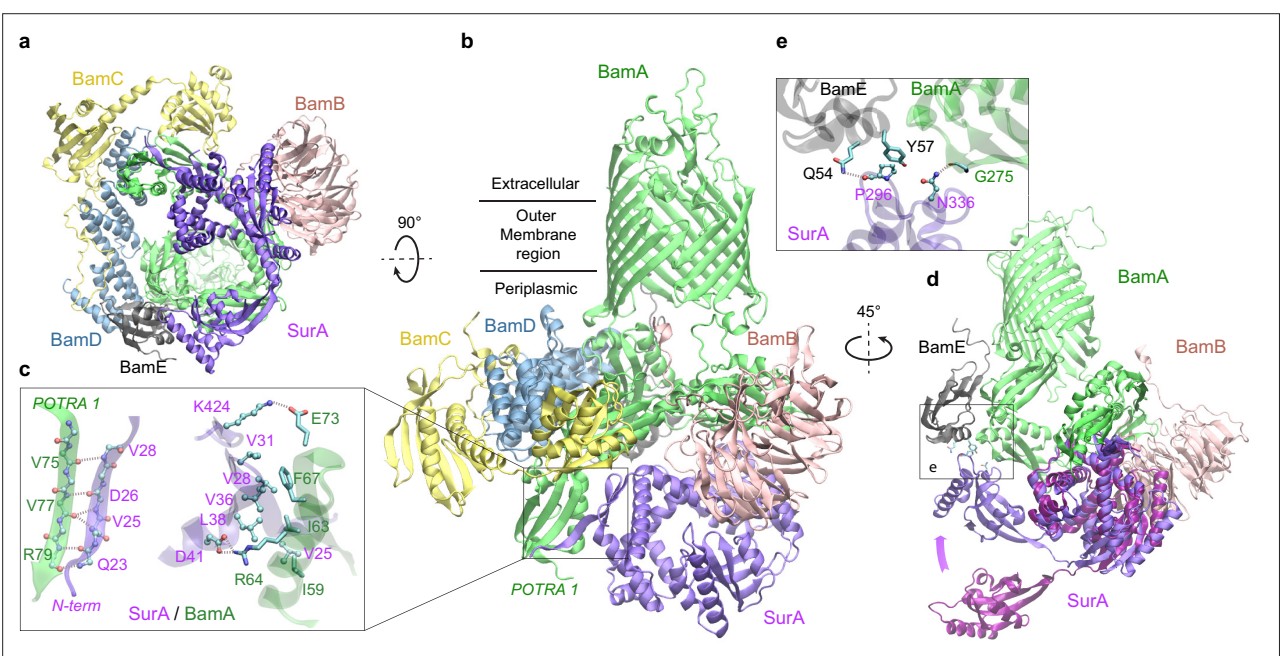

**Figure 5.** Structural model of SurA docked to β-barrel assembly machine (BAM). (**a** and **b**) Two views of the top ranked supercomplex model in the cartoon representation. The BAM constituents are BamA (green), BamB (pink), BamC (yellow), BamD (blue), and BamE (black). The N-terminal POTRA1 domain of BamA provides the main interaction sites for SurA (violet). (**c**) Close-up view of the interaction sites at POTRA1 and the core domain of SurA. Interacting residues are shown in the licorice (SurA) and ball-and-stick (BamA) representations. (**d**) Crystal structure of SurA (magenta, PDB 1M5Y) superimposed onto the computed structure of the supercomplex. The magenta arrow indicates the change of location in P2 between two structures. BamCD are omitted for clarity. (**e**) Protein-protein interaction between P2 of SurA and POTRA4 of BamA and BamE.

four-prong hook, provides SurA with another configuration to handle clients, potentially large ones, consistent with experimental conclusions (*Li et al., 2018*; *Calabrese et al., 2020*).

## SurA and BamA specifically recognize each other

Excitingly, envelopome screening also confidently detected that SurA interacts with BamA, the anchoring subunit of BAM at the final stop of OMPs. To understand how SurA interacts with BAM, we subsequently used AF2Complex to probe potential interactions of SurA with all five BAM constituents (BamABCDE) and acquired a high confidence complex structure (iScore = 0.75, mean pLDDT = 84; *Figure 5a–b*). Because BAM has been extensively studied structurally (*Tomasek and Kahne, 2021*; *Wu et al., 2020*), we focus on describing its interaction with SurA, though the predicted BAM structure closely mimics a known crystal structure of the complex determined at 2.9 Å resolution (PDB 5D0O; *Gu et al., 2016*). The alignment of the two complex structures yielding a very high TM-score of 0.94. Overall, SurA mainly interacts with BamA, with similar interactions observed in both the top supercomplex and the BamA/SurA dimer structures, in the N-terminal domains of both proteins (*Figure 5b–c*). $Gln23_{SurA}$ to $Val28_{SurA}$, largely missing in the crystal structures of SurA (*Bitto and McKay, 2002*; *Xu et al., 2007*), now form a β-sheet with the β2 strand of POTRA1, the N-terminal domain of BamA. In addition, hydrophobic contacts and two salt bridges are present in the same region. The structure explains the result that $Asp26_{SurA}$ is cross-linked to BamA (*Wang et al., 2016*) because $Asp26_{SurA}$ is located at the center of the SurA/BamA interface. Furthermore, SurA adopts a closed conformation in our model, but the P2 domain rotates ~45° from its crystal position (*Bitto and McKay, 2002*) to engage additional contacts with BamA and BamE (*Figure 5d–e*). For example, $Asn336_{SurA}$ forms a hydrogen bond with the backbone oxygen atom of $Gly275_{BamA}$; in the same cross-linking study

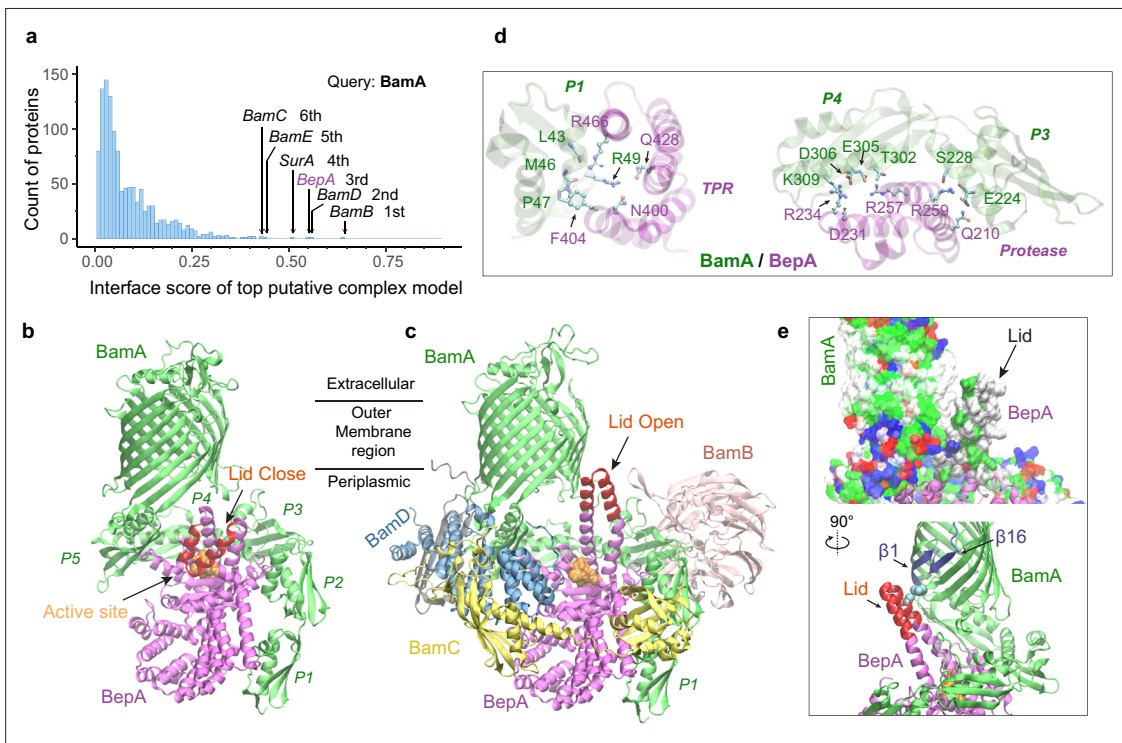

**Figure 6.** Structural models of β-barrel assembly machine (BAM) and BepA. (**a**) Computational protein-protein interaction screening identifies BepA as a top hit to BamA. (**b**) Top structural model of the heterodimeric complex of BamA (green) and BepA (purple) in cartoon representation. The lid of BepA is colored red. The active sites with the protease domain of BepA are shown in a surface representation (orange). The five POTRA domains of BamA are labeled P1–P5. (**c**) Predicted structure of the BAM/BepA supercomplex. The lid of BepA extends to an open conformation. The image was created from the same viewpoint as **b**. (**d**) Specific residue-residue contacts between BamA and the tetratricopeptide repeat (TPR) and protease domains of BepA. (**e**) Close-up views of the lid and the BamA β-barrel in the surface (top) and cartoon representations (bottom). A hydrophobic contact between $Ala180_{BepA}$ and $Leu780_{BamA}$ is shown as spheres, and the lateral gate of BamA is between the β1 and β16 strands (dark blue).

The online version of this article includes the following figure supplement(s) for figure 6:

**Figure supplement 1.** Predicted structures of BepA compared to two experimental structures.

(*Wang et al., 2016*), Asn336$_{SurA}$ is implied to interact with BamA. These results invite a hypothesis that SurA and BamA initiate payload transfer via specific docking at their N-terminal domains.

## Metalloprotease BepA flips a lid in complex with BAM

SurA led us to BamA, for which we conducted another envelopome PPI screening. The top six hits include all expected ones, the other four BAM factors and SurA, but unexpectedly, BepA ranked third with high confidence (*Figure 6a–b*). BepA was not anticipated because we were not aware of its relevance. However, a literature search quickly revealed that BepA is highly relevant because it cleans up stalled OMP folding at BAM (*Narita et al., 2013*; *Soltes et al., 2017*), and its interactions with BAM have been documented (*Narita et al., 2013*; *Daimon et al., 2017*), though no structure of the complex is available. Consequently, we modeled BepA and the full BAM complex altogether, obtaining a high confidence model (iScore = 0.68, mean pLDDT = 85, *Figure 6c*). Overall, the BepA/BamA interfaces are quite similar (interface C$_\alpha$ RMSD 1.2 Å) between the dimer and the supercomplex structure. Extensive contacts are present between all five periplasmic domains (POTRA1−5) of BamA and the two domains (protease and TPR) of BepA. They can be clustered into two main groups scattered between the POTRA1 and TPR and between POTRA3-4 and the protease domains (*Figure 6d*). Strong photo cross-linking signals to BamA have been observed previously for several residues of TPR including Phe404$_{BepA}$ (*Narita et al., 2013*; *Daimon et al., 2017*). According to our model, Phe404$_{BepA}$ belongs to the first interaction cluster with five BamA residues, particularly Pro47$_{BamA}$. Another residue cross-linked to BamA is Gln428$_{BepA}$, which establishes a specific hydrogen bond with Arg49$_{BamA}$ in our model.

Crystal structures of BepA alone have been solved, but a segment (residue 154–192) of the metalloprotease domain is absent in these structures, presumably due to its flexibility (*Daimon et al., 2017*; *Bryant et al., 2020*; *Shahrizal et al., 2019*; *Figure 6—figure supplement 1*). The segment, termed an 'active-site lid,', was speculated to be important to the protease function of BepA because its movement could either expose (open state) or cover (closed state) the catalytic site, the HEXXH motif (*Bryant et al., 2020*). Intriguingly, we only see the closed conformation in 40 models of the BamA/BepA heterodimer (*Figure 6b*), but both the open and closed states in 8 supercomplex models. In the open state, two α-helices comprising the lid are in a straight-up configuration (*Figure 6c*), whose upper half (residue 157–186) rotates ~105° to cover the enzymatic site in the closed configuration (*Figure 6b*). Moreover, the flexible lid, with a large hydrophobic surface similar to the β-barrel of BamA, is in a position as if it is inserted into the inner leaflet of the OM and is in the proximity of the lateral gate of BamA (*Figure 6e*). As such, BepA could control its protease activity via probing a substrate β-barrel stemming out of BamA. Hence, it is hypothesized that a normal budding β-barrel blocks the lid opening, whereas a stalled OMP within BamA permits lid opening and subsequent substrate cleavage. It is also possible that the lid of BepA somehow detects an abnormal β-barrel popping out of BamA, as suggested in an experimental study that shows BepA degrades a LptD mutant stalled at a late stage of β-barrel folding (*Soltes et al., 2017*).

## Discussion

Here, we have demonstrated a deep learning strategy that combines virtual PPI screening over the *E. coli* envelopome and supercomplex structure modeling. By applying it to several key proteins in the OMP biogenesis pathway, we have identified their functional partners within the top 1% ranking of ~1450 proteins screened for PPIs per query. Thanks to high confidence structures underlying the top predictions, one can understand many experimental phenomena, particularly in vivo site-directed photo cross-linking data. For example, cross-linked products found from the SecYEG or BAM supercomplexes may be explained by direct physical interactions revealed in our predicted structures. Moreover, previously speculated conformations are captured for SurA and BepA. Most importantly, these revealing atomic structures suggest mechanistic hypotheses for various steps of the OMP biogenesis pathway as summarized in *Figure 7*, where we present their diagrams along with some predicted supercomplex structures.

One unexpected discovery is the DsbA/PpiD interaction. It was known that DsbA crucially transfers its disulfide bond to a nascent polypeptide translocated by SecYEG (*Kadokura and Beckwith, 2009*; *Goemans et al., 2014*), but how does the translocon interact with DsbA? The predicted supercomplex

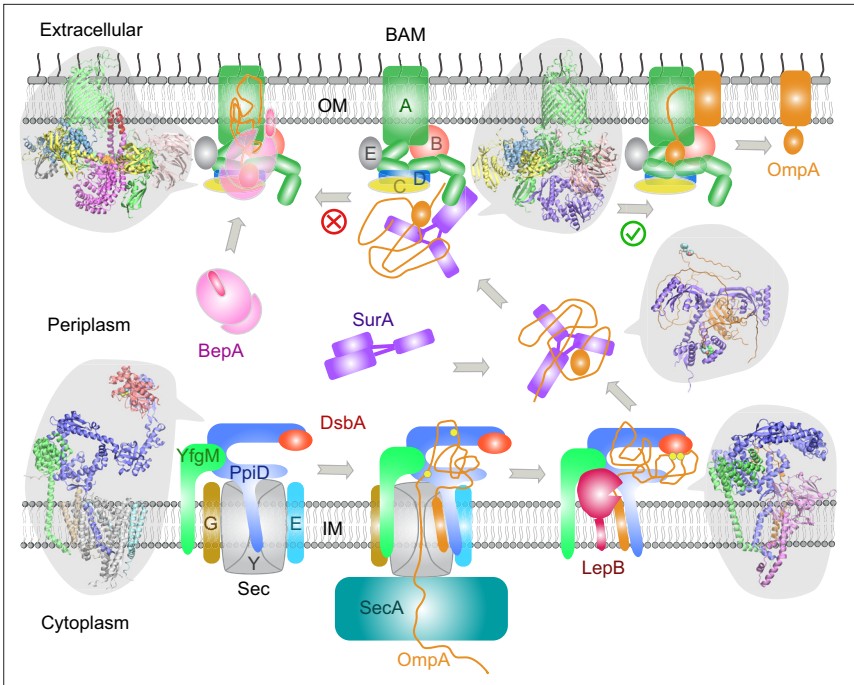

**Figure 7.** Proposed mechanisms involved in the outer membrane protein (OMP) biogenesis pathway in *E. coli*. Complex structures resulting from this study accompany relevant cartoon diagrams. Powered by SecA, a precursor OmpA polypeptide (orange line) first passes through the SecYEG translocon in complex with PpiD and YfgM. PpiD, held in place by YfgM, senses the translocating substrate via its N-terminal α-helix bound to the lateral gate of SecY and temporarily dissociates from the translocon upon receiving the substrate OmpA. Protein disulfide isomerase DsbA is recruited by PpiD and promotes formation of a disulfide bond between two cysteine residues (yellow spheres) of OmpA. Meanwhile, peptidase LepB fills the vacancy left by SecYEG and cleaves the transmembrane signal peptide from OmpA, which is then handed over to chaperone SurA. At this point, the periplasmic domain of OmpA is folded, but the unfolded β-barrel region wraps around SurA, which carries OmpA to BAM. Lastly, SurA docks to POTRA1, the N-terminal domain of BamA, where the β-barrel domain of OmpA is folded and released from the lateral gate of BamA. If this folding and assembly process are stalled for some reason, metalloprotease BepA senses the failure with its flexible lid and cleans up by cleaving a stuck substrate. For clarity, the peptidoglycan layer in the periplasm is not shown, and the schematic drawings are not to scale.

of SecYEG/PpiD/YfgM/DsbA provides a compelling answer for three reasons: first, the transient residence of DsbA on PpiD at the translocon dramatically improves its efficiency versus a random search in a crowded periplasm. Second, it predicts a function for the PPIase domain of PpiD that mysteriously lacks the PPIase function. Third, since PpiD is anchored to the IM, it can keep DsbA close to DsbB, an IM protein that recycles DsbA. Notably, two other *E. coli* chaperones, SurA and FkpA, possess two PPIase domains each, and AF2Complex predicts that DsbA interacts with FkpA (iScore = 0.47 versus 0.55 for PpiD) but not with SurA (iScore = 0.03).

Interestingly, the AF2 neural networks can model a partially unfolded polypeptide accompanied by chaperones, even though AF2 was trained on folded proteins. We attained structural models that appear to be at least partially physical. One example is the model of PpiD/YfgM/LepB/proOmpA, in which proOmpA is posed for cleavage by peptidase LepB. The cleavage alanine of OmpA is ~4 Å away from the catalytic triad of LepB. Considering that only apo or inhibitor bound structures of LepB are available, this computed model implies that AF2 has learned physical representations. More intriguing examples are the models of SurA/OmpA, where the periplasmic domain of OmpA is folded, but the β-barrel domain is completely unfolded and loosely wrapped around SurA. The predicted structures echo the NMR structures of an unfolded polypeptide bound to SecB, a cytosolic chaperone involved in the early stage of the SecA-dependent translocation pathway (*Huang et al., 2016*). Despite the low confidence due to weak interactions, the predicted structures delineate a picture for how SurA prevents OmpA from aggregating. Moreover, since it transports OmpA with a relatively small number of intermolecular contacts, the free energy required to dissociate OmpA from SurA is

small. Notwithstanding these considerations, we caution that artifacts likely exist in these predicted structural models.

These results reinforce the notion that deep learning is a promising way to explore the conformational ensemble of proteins and to uncover the molecular mechanisms of biosystems (*Gao et al., 2022b*). The combination of advanced AF2 deep learning models, an effective PPI ranking metric, and a workflow optimized for large-scale screening generates illuminating structures of protein complexes. In the presented examples, we focused on those with obvious biological relevance according to the literature. There are of course other confident PPI predictions that were not described here. While some are promiscuous interactions or physically possible but biologically irrelevant, or simply false, there are likely functional interactions yet to be explored. More generally, these results are an example of a deep learning-based strategy that helps to elucidate mechanistic aspects of complex biochemical pathways.

## Methods
### *E. coli* cell envelopome

The proteome for *E. coli* strain K12 MG1655, consisting of 4400 protein sequences, was retrieved from UniProt (*Wu et al., 2006*) in March 2022 (Proteome ID UP000000625). Then, the subcellular location of each protein was parsed to collect all known and predicted cell envelope proteins. To be conservative, we included all proteins whose primary subcellular location is not the cytosol, yielding 1466 proteins defined as the envelopome. The set contains all proteins located in the IM, periplasm, OM, and extracellular surface, and some cytosolic proteins located primarily at the periphery of the periplasmic IM, e.g., SecA. For AF2Complex modeling, we removed 11 sequences that contain at least one non-standard amino acid.

### AF2Complex

An updated version of AF2Complex was built upon AlphaFold version 2.2.0 (*Evans et al., 2021*). AF2Complex supports three different set of deep learning neural network models provided with AF2, i.e., the original models for monomer prediction (AF version 2.0.1 *Jumper et al., 2021*) and two set of models for multimer prediction (version 2.1.0 and version 2.2.0; *Evans et al., 2021*). Numerous changes were made; we list three major ones here. First, the iScore metric was introduced to rank the confidence of a predicted complex model (*Gao et al., 2022b*). The iScore metric was derived from the predicted alignment errors that give an estimated distance for interface residue $j$ from its position in the experimental structure, as viewed from a local frame of residue interface residue $i$ (*Gao et al., 2022b*; *Jumper et al., 2021*). To better estimate confidence, the contribution of each interface residue to the iScore is calculated using local frames not located within the same protein chain, i.e., residue $i$ and $j$ belonging to different chains. Second, the data pipeline for generating input features was split from the neural network inference. This allows rapid assembly of input features for individual proteins of a putative complex from pre-generated input features of the full proteome. Third, five options for multiple sequence alignment (MSA) pairing are provided: no pairing, all paired, cyclic, linear, and arbitrary pairing. The last three options are experimental, and we only employed the first two MSA pairing modes in this work. All structural models, either of a monomer or a multimer, were predicted by AF2Complex using either the '*monomer_ptm*' deep learning models without any MSA pairing or the '*multimer_v2*' models with all MSA pairing; the latter only used for structure prediction of complexes. The input features of all monomers were derived using standard sequence libraries (*The Uniprot Consortium, 2019*; *Mitchell et al., 2020*; *Tunyasuvunakool et al., 2021*) and a version of the Protein Data Bank (*Berman et al., 2000*) released in November 2021. The confidence of an output structure is evaluated using three different metrics: the iScore for protein interface (*Gao et al., 2022b*), the predicted TM-score for global structure of a monomer (*Zhang and Skolnick, 2004*), and the pLDDT score for local domain structure (*Jumper et al., 2021*; *Mariani et al., 2013*). According to AF2, a mean pLDDT score higher than 70 indicates high confidence in a predicted structure when evaluated on individual domain(s).

## Envelopome PPI screening

We adapted a workflow for proteome-scale monomeric structure prediction with AF2 implemented on the Summit supercomputer at Oak Ridge National Laboratory (*Gao et al., 2022a*). The workflow was applied to four query proteins: PpiD, YfgM, SurA, and BamA. Given a query sequence and an envelopome protein sequence, AF2Complex assembles the pre-generated monomeric input features and then feeds the composite features to the AF2 neural network models for inference. 10 different neural network models are used as mentioned above, and each deep learning model is invoked twice with two different random seeds and up to eight recycles. This procedure typically gives 20 final structural models per target for ranking, and the top ranked model by iScore is retained. Although the '*multimer_v2*' set of models greatly reduces unphysical clashes in predicted structures compared to structures predicted by the original set of multimer deep learning models, there is still a small chance of generating severe clashes, which could yield an artifactually high iScore. We filter out these structures by applying a minimum interface clash indicator of 0.4 (*Gao et al., 2022b*). We did not apply model relaxation to these predicted structures because most are not actual protein complexes; as such, it is not worth expending the extra computing time. To avoid memory overflow, we limited the input MSA depth to 5000 for each monomer and allowed a maximum of four structural templates per monomer. A maximum of 1600 amino acids was imposed on the total size of each complex target. This limitation slightly reduced the total number proteins in the envelopome for screening, typically to ~1450 for the four proteins studied. Because about 30% of envelopome proteins have a signal peptide that is absent in their mature chains, only mature chains were used for screening. This is readily realized with AF2Complex because it provides an option to crop arbitrary segments of a monomer input feature during feature assembly. The residue ranges of the mature chains were obtained from the UniProt knowledgebase.

## Modeling the OmpA polypeptide

It was necessary to model OmpA as a substrate of PpiD or SurA. OmpA was chosen mainly because it is a model OMP for studying the OMP biogenesis pathway (*Reusch, 2012*). Many experimental data exist for validation. To minimize potential 'memory' effects due to large MSAs or structural templates, we reduced the number of sequences in the MSAs and removed all structural templates in the input features of OmpA. We tested MSA depths of 1, 10, 20, 50, and 100 to predict a structural model of proOmpA in complex with SecYEG. The goal is to generate a structural model that mimics the crystal structure of SecYEG translocating a proOmpA polypeptide (*Li et al., 2016*). Our tests found that an MSA depth below 50 can yield a model structure of an unfolded OmpA peptide through the channel of SecY, when used with either '*model_1_ptm*' or '*model_3_ptm*' of all AF2 neural network models. In this study, we used an MSA depth of 20 and the 2 AF2 models to predict structural models involving OmpA.

## Computational resources

*E. coli* envelopome PPI screening was performed on the Summit supercomputer, typically using 923 nodes for several hours of wall clock time. Each Summit node hosts six Nvidia 16 GB V100 GPUs. The structure predictions of various supercomplexes were conducted locally using about 10 workstations each with four Nvidia RTX6000 GPUs, each with 24 GB of GPU memory.

## Analysis

The program VMD (*Humphrey et al., 1996*) was used to inspect predicted structural models and create all molecular images. APoc was used to align monomeric protein structures (*Gao and Skolnick, 2013*), and iAlign was used to perform protein-protein interface comparison (*Gao and Skolnick, 2010*).

## Acknowledgements

We thank Jerry M Parks for stimulating discussions and critical reading of the manuscript. This work was supported in part by the DOE Office of Science, Office of Biological and Environmental Research (DOE DE-SC0021303) and the Division of General Medical Sciences of the National Institutes Health (NIH R35GM118039). The research used resources supported in part by the Advanced Scientific

Computing Research (ASCR) Leadership Computing Challenge (ALCC) program, and by the Partnership for an Advanced Computing Environment (PACE) at the Georgia Institute of Technology.

## Additional information

### Funding

| Funder | Grant reference number | Author |
|---|---|---|
| U.S. Department of Energy | DE-SC0021303 | Jeffrey Skolnick |
| National Institute of General Medical Sciences | R35GM118039 | Jeffrey Skolnick |
| Advanced Scientific Computing Research | | Mu Gao<br>Jeffrey Skolnick |
| Advanced Computing Environment (PACE) | | Jeffrey Skolnick |

The funders had no role in study design, data collection and interpretation, or the decision to submit the work for publication.

### Author contributions

Mu Gao, Conceptualization, Resources, Data curation, Software, Formal analysis, Supervision, Funding acquisition, Investigation, Visualization, Methodology, Writing - original draft, Writing - review and editing; Davi Nakajima An, Software, Methodology, Writing - review and editing; Jeffrey Skolnick, Conceptualization, Resources, Supervision, Funding acquisition, Project administration, Writing - review and editing

### Author ORCIDs

Mu Gao http://orcid.org/0000-0002-0378-3704
Jeffrey Skolnick http://orcid.org/0000-0002-1877-4958

### Decision letter and Author response

Decision letter https://doi.org/10.7554/eLife.82885.sa1
Author response https://doi.org/10.7554/eLife.82885.sa2

## Additional files

### Supplementary files

• Supplementary file 1. Top hits from the protein-protein interaction (PPI) screening over the *E. coli* envelopome with AF2Complex for query proteins: PpiD, YfgM, SurA, and BamA.

• MDAR checklist

### Data availability

The input features for the full *E. coli* proteome used for envelopome screening by AF2Complex, and the computational models presented in this study are available at Zenodo (https://doi.org/10.5281/zenodo.6846915). The source code of AF2Complex is freely available at https://github.com/FreshAirTonight/af2complex.

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
