## [Editor Report]

The authors show that an artificial-intelligence method can be used to predict the three-dimensional structure of protein-protein complexes formed between cellular factors that promote the assembly of bacterial outer membrane proteins. The structures are compelling because they explain previously published biochemical data and provide novel insights into the function of these factors.

---

## [Decision Letter]

**Decision letter after peer review:**

Thank you for submitting your article "Deep learning-driven insights into super protein complexes for outer membrane protein biogenesis in bacteria" for consideration by *eLife*. Your article has been reviewed by 3 peer reviewers, overseen by José Faraldo-Gómez as the Senior Editor. Two of the reviewers have agreed to reveal their identity, namely Nir Ben-Tal (Reviewer #1); Arne Elofsson (Reviewer #3).

As you will see below, the reviewers conclude the manuscript is not suitable for publication for *eLife* in its current form, but they make specific recommendations to resolve their concerns. I recognize some of these revisions are substantial – nevertheless, I encourage you to take the time required to address these concerns convincingly and then submit a revised version.

*Reviewer #2 (Recommendations for the authors):*

In this manuscript, Gao et al., combine a very elegant virtual screening method (AF2Complex) based on Α Fold with known crystal structures to predict the structure of super-protein complexes in the *E. coli* cell envelope that play important roles in the biogenesis of outer membrane proteins (OMPs). The authors do an excellent job of validating their methodology. Their predicted structures both explain a considerable amount of biochemical data that has appeared in the literature (e.g., the crosslinking of PpiD to SecY and SurA to BamA) and provide compelling explanations for unresolved conceptual issues (e.g., how DsbA is recruited to the Sec machinery to form disulfide bonds as proteins are translocated into the periplasm). With one exception (see comment 4 below), the predicted structures also provide potentially important insights into the function of the assembly factors. In essence, this work describes an impressive example of the power of AI that should be of interest not only to investigators who study OMP biogenesis but also to the broader research community. I should also note that the manuscript is clearly and concisely written for a general audience.

Specific comments:

1) Lines 83-86 and Figures 1a, 4a and 6a: The authors selected predicted structures of known biological relevance for further study, but they do not comment on high confidence predictions of supercomplex structures of unknown biological significance. How can they distinguish predicted structures that have potential biological significance from those that simply represent noise/false positives? This is an important issue that requires further explanation, especially because the authors provide false-positive rates on lines 76/77.

2) In the paper by Alvira et al., (ref. 21) the authors argue that *E. coli* produce a "holotranslocon" in which SecDF and BAM interact across the periplasm. I was surprised that AF2Complex did not detect this interaction with a high confidence score. Does AF2Complex have a significant rate of "false negatives"? The authors should discuss this issue.

3) Lines 181-183 and Figure 4c: As suggested by many different studies, the C-terminal β signal of OmpA must be accessible to bind to the first β strand of BamA at a relatively early stage of assembly. Would the last β strand be accessible if the β barrel is wrapped around SurA? The authors should comment on this issue.

4) Lines 246-249: While the structural model of the BAM-BepA supercomplex shown in Figure 6 is intriguing, the idea that the protease activity of BepA is controlled by the probing of substrate β barrels as they bud from the BamA β barrel does not appear to be consistent with the literature. The authors propose that β barrels that are stalled within BamA permit the opening of the BepA lid and activation of the protease. This suggests that BepA degrades β barrels that are stalled at an early stage of assembly, but not at later stages. The work of Soltes, et al., (ref. 48), however, shows that BepA degrades an LptD mutant that has formed a nearly complete barrel (and that likely protrudes from the BamA barrel) while another protease (YcaL) degrades a second LptD mutant that engages BAM at an earlier stage of assembly. I realize that the stages of OMP assembly are currently rather vague, but the authors should discuss the work of Soltes et al., and state that it imposes a possible caveat on their model.

*Reviewer #3 (Recommendations for the authors):*

1) What is the advantage of running both monomer and multimer versions of AF2? Does it provide an advantage? In our benchmarks, the multimer is slightly better and roughly doubles the computational cost. Also, did you run 2 multimer versions (version 2.1.0 should not be used it makes bad predictions often)?

2) Why is not the top-hit for surA (oppA) discussed? Or any other of the high-scoring pairs?

3) The authors claim that an iScore of 0.4 is related to a 1.2% FPR. This would mean that yfgM would have roughly 100 interactions. Is this correct? If so, this would need to be discussed (and possibly proven experimentally). Here only discussion with PPID at rank 12 is discussed. Should it not be more likely that the other 11 higher-ranked models interact? Why are these ignored?

4) Would it not be computationally more efficient to use MMseqs2 to make the MSAs?

5) Why is it only tested on 4 proteins? I think it should be computationally possible to run it on many more (in particular as you already have the MSAs). At a minimum, all complexes shown in Figure 7 should be run through the pipeline to ensure these can be modelled.

6) How would the pipeline handle stoichiometry?

7) WHat happens if a set of the high-scoring pairwise interactions are fed into alphafold-multimer (which can handle up ~5000 residues). Can the large complexes be modelled?

---

## [Author Response]

Reviewer #2 (Recommendations for the authors):1) Lines 83-86 and Figures 1a, 4a and 6a: The authors selected predicted structures of known biological relevance for further study, but they do not comment on high confidence predictions of supercomplex structures of unknown biological significance. How can they distinguish predicted structures that have potential biological significance from those that simply represent noise/false positives? This is an important issue that requires further explanation, especially because the authors provide false-positive rates on lines 76/77.

We would like to clarify that a true positive of our method is about predicting physical interactions between the two proteins subjected to modeling alone. It does *not* consider many other factors such as stoichiometry, binding competition with other proteins, post-translational modifications, and functional relevance. As a result, we must review other sources of evidence and select ones that likely have biological relevance according to existing literature. Ideally, such a selection requires the involvement of experts of these proteins or pathways. For this reason, we provide the lists of all top predictions in case that the hits we ignored could turn out to be useful for future studies by the respective experts.

2) In the paper by Alvira et al., (ref. 21) the authors argue that *E. coli* produce a "holotranslocon" in which SecDF and BAM interact across the periplasm. I was surprised that AF2Complex did not detect this interaction with a high confidence score. Does AF2Complex have a significant rate of "false negatives"? The authors should discuss this issue.

The paper referred to by the reviewer is one of several intriguing works that led us to this study. While we were able to obtain high confidence models for some complexes found in previous experimental studies, in the case of BamA/SecDF, unfortunately we did not observe a confident computational model, despite multiple run efforts. In retrospect, it is unclear to us whether or not direct interactions between BamA and SecDF alone would be feasible in a normal periplasmic space, which is about 20 nm wide from the inner membrane to outer membrane. Having said that, it is entirely possible that our computational method missed the target as a false negative, as the reviewer speculated. In this revision, we have added statistics of the recall (1 – false negative rate), from line 77,

“With respect to the capability of identifying and modeling true protein-protein interactions, on a set of 440 heterodimeric complexes whose experimental structures have been recently determined and were not used for AF2 model training, AF2Complex recalls 81%, 74% and 34% of these benchmark targets at the same three iScore thresholds, respectively, and yields medium- or high-quality complex structures for 84%, 87%, and 93% in the top ranked model of these positively identified targets [23].”

3) Lines 181-183 and Figure 4c: As suggested by many different studies, the C-terminal β signal of OmpA must be accessible to bind to the first β strand of BamA at a relatively early stage of assembly. Would the last β strand be accessible if the β barrel is wrapped around SurA? The authors should comment on this issue.

We thank the reviewer for this suggestion. We have verified that the so-called β-signal residues are accessible in our OmpA/SurA complex models.

We marked the two referred to residues in the revised Figure 4 and added the following sentence in line 211,

“Of note, two key ‘β-signal’ residues [47], Y189 and F191 of OmpA, do not make direct physical contact with SurA in both models.”

4) Lines 246-249: While the structural model of the BAM-BepA supercomplex shown in Figure 6 is intriguing, the idea that the protease activity of BepA is controlled by the probing of substrate β barrels as they bud from the BamA β barrel does not appear to be consistent with the literature. The authors propose that β barrels that are stalled within BamA permit the opening of the BepA lid and activation of the protease. This suggests that BepA degrades β barrels that are stalled at an early stage of assembly, but not at later stages. The work of Soltes, et al., (ref. 48), however, shows that BepA degrades an LptD mutant that has formed a nearly complete barrel (and that likely protrudes from the BamA barrel) while another protease (YcaL) degrades a second LptD mutant that engages BAM at an earlier stage of assembly. I realize that the stages of OMP assembly are currently rather vague, but the authors should discuss the work of Soltes et al., and state that it imposes a possible caveat on their model.

We thank the reviewer for pointing out these experimental observations that we had inadvertently overlooked. Our hypothesis is a quite simplified reasoning based on computational models and some experimental studies that we were aware of at the time we wrote the original version of this paper. We have now revised the hypothesis and added an alternative possibility on line 271,

“It is also possible that the lid of BepA somehow detects an abnormal β-barrel popping out of BamA, as suggested in an experimental study that shows BepA degrades a LptD mutant stalled at a late stage of β-barrel folding [51].”

Reviewer #3 (Recommendations for the authors):1) What is the advantage of running both monomer and multimer versions of AF2? Does it provide an advantage? In our benchmarks, the multimer is slightly better and roughly doubles the computational cost. Also, did you run 2 multimer versions (version 2.1.0 should not be used it makes bad predictions often)?

We agree with the reviewer that the multimer_v2 deep learning models by DeepMind (AF version 2.1.0) provide improved accuracy in predicted models, and we employed this version of multimer models instead of the multimer_v1 version. However, in our benchmarks, we found that the original monomer deep learning models (AF version 2.0.1) make good predictions not seen with the multimer models in some cases. For example, as we mentioned in the Methods (line 382), only certain monomer deep learning models could generate models of an unfolded OmpA peptide through the channel of SecY mimicking the crystal structure (Ref. 12). For this reason, we included the monomer models in our procedure as well. This practice slightly improves the coverage of our approach.

2) Why is not the top-hit for surA (oppA) discussed? Or any other of the high-scoring pairs?3) The authors claim that an iScore of 0.4 is related to a 1.2% FPR. This would mean that yfgM would have roughly 100 interactions. Is this correct? If so, this would need to be discussed (and possibly proven experimentally). Here only discussion with PPID at rank 12 is discussed. Should it not be more likely that the other 11 higher-ranked models interact? Why are these ignored?

We thank the reviewer for these good questions, which we have addressed in R1.2 as they were raised similarly by the other two reviewers. We again stress that our selections from predicted interacting protein pairs were based on biological relevance that we were aware of. With respect to the false positive rate, it was based on benchmarking a set of putatively non-interacting protein pairs in *E. coli* (Ref 23). It is possible that some of these false positives with high scores could physically interact if the proteins were actually present at the same subcellular location, but in the cell they cannot as they are located in different cellular compartments, one main criterion employed for selecting these control negatives. Obviously, such interacting pairs are unlikely to be biologically relevant. Moreover, even though we screened about 1,500 envelope proteins, the numbers of confident hits are generally low, e.g., 9 for BamA, 12 for PpiD, and it is clear that at least some of these hits are not false positives. The query protein with highest number of confident hits is YfgM, which has 98 hits above an iScore of 0.4. They are all attracted to the plam-like surface of YfgM (Figure 1c), the same surface where PpiD is predicted to make extensive contacts with YfgM. Thus, PpiD likely forms a very stable complex with YfgM that other hits may not be able to compete with. Therefore, the biological relevance of other YfgM hits is unclear. We skipped all hits that we could not convince ourselves by some molecular hypothesis that they are relevant to the OMP biogenesis pathway.

4) Would it not be computationally more efficient to use MMseqs2 to make the MSAs?

We cannot comment on the efficiency of MMseqs2 because we have not benchmarked its performance. However, as mentioned in the Methods, we have implemented a workflow in AF2Complex such that it can re-use the input features including the MSAs of monomers for multimer predictions. This means that one needs to generate MSAs only once for each protein in a proteome and reassemble them quickly for predicting arbitrary combinations of multimers. This eliminates the need for feature derivation for each multimer target as is required in DeepMind’s workflow, which is far less efficient, especially for large-scale screening. We have shared the input features of the full *E. coli* proteome at Zenodo (see Data availability), and we hope that they could be helpful for modeling other *E. coli* protein complexes.

5) Why is it only tested on 4 proteins? I think it should be computationally possible to run it on many more (in particular as you already have the MSAs). At a minimum, all complexes shown in Figure 7 should be run through the pipeline to ensure these can be modelled.

All complexes presented in Figure 7 actually resulted from the modeling efforts of this study, though parts of these complexes were experimentally determined in previous studies. We wish that we could conduct all-against-all PPI screening of *E. coli*., but we could not afford the huge computational costs to do such a study. Another limiting factor is the brain power required for analyzing the results, as we have neither the resources nor the expertise to analyze hundreds or thousands of PPI predictions by ourselves. The main purpose of the study is to demonstrate the usefulness of a new approach to the study of biological pathways through what we believe biologically relevant and important examples.

6) How would the pipeline handle stoichiometry?

The pipeline requires a user to specify a stoichiometry of a complex target. In this study, we rely on the literature to determine the stoichiometry. If such information is not available, one could run some trials at different stoichiometries and select the stoichiometry that gives the most confident model. While this is possible in some cases, it is in general a challenging issue.

7) WHat happens if a set of the high-scoring pairwise interactions are fed into alphafold-multimer (which can handle up ~5000 residues). Can the large complexes be modelled?

As we show for the BAM/BepA complex, it is possible to make confident models for large complexes that are made of multiple high-scoring pairwise interactions. However, it is also our experience that large scale modeling is challenging when the total size of the complex is over 3000 residues. This is very much a frontier research topic that requires further exploration.